# Conditional Probability of Survival and Prognostic Factors in Long-Term Survivors of High-Grade Serous Ovarian Cancer

**DOI:** 10.3390/cancers12082184

**Published:** 2020-08-05

**Authors:** Michel Fabbro, Pierre-Emmanuel Colombo, Cristina Marinella Leaha, Philippe Rouanet, Sébastien Carrère, François Quenet, Marian Gutowski, Anne Mourregot, Véronique D’Hondt, Isabelle Coupier, Julie Vendrell, Paul Vilquin, Pascal Pujol, Jérôme Solassol, Caroline Mollevi

**Affiliations:** 1Medical Oncology Department, Montpellier Cancer Institute (ICM), University of Montpellier, 3429834090 Montpellier, France; Veronique.Dhondt@icm.unicancer.fr; 2Surgical Oncology Department, Montpellier Cancer Institute (ICM), University of Montpellier, 34298 Montpellier, France; Pierre-Emmanuel.Colombo@icm.unicancer.fr (P.-E.C.); Philippe.Rouanet@icm.unicancer.fr (P.R.); sebastien.carrere@icm.unicancer.fr (S.C.); francois.quenet@icm.unicancer.fr (F.Q.); marian.gutowski@icm.unicancer.fr (M.G.); anne.mourregot@icm.unicancer.fr (A.M.); 3Pathological Department, Montpellier Cancer Institute (ICM), University of Montpellier, 34298 Montpellier, France; Cristina.Leaha@icm.unicancer.fr; 4Oncogenetics Department, Montpellier Hospital University, University of Montpellier, 34298 Montpellier, France; isabelle.coupier@icm.unicancer.fr (I.C.); p-pujol@chu-montpellier.fr (P.P.); 5Solid Tumors Biology Department, Montpellier Hospital University, University of Montpellier, 34298 Montpellier, France; j-vendrell@chu-montpellier.fr (J.V.); paul.vilquin@aphp.fr (P.V.); j-solassol@chu-montpellier.fr (J.S.); 6Biometrics Unit, Montpellier Cancer Institute (ICM), University of Montpellier, 34298 Montpellier, France; Caroline.Mollevi@icm.unicancer.fr

**Keywords:** high-grade serous ovarian cancer, conditional probability of survival, long-term survival, prognostic factors

## Abstract

*Objective:* High-grade serous ovarian cancers (HGSOC) are heterogeneous, often diagnosed at an advanced stage, and associated with poor overall survival (OS, 39% at five years). There are few data about the prognostic factors of late relapses in HGSOC patients who survived ≥five years, long-term survivors (LTS). The aim of our study is to assess the probability of survival according to the already survived time from diagnosis. *Methods:* Data from HGSOC patients treated between 1995 and 2016 were retrospectively collected to estimate the conditional probability of survival (CPS), probability of surviving Y years after diagnosis when the patient had already survived X years, and to determine the LTS prognostic factors. The primary endpoint was OS. *Results:* 404 patients were included; 120 of them were LTS. Patients were aged 61 years (range: 20–89), WHO performance status 0–1 in 86.9% and 2 in 13.1%, and Fédération Internationale de Gynécologie et d’Obstétrique (FIGO) staging III and IV in 82.7% and 17.3% patients. Breast cancer (BRCA) status was available in 116 patients (33% mutated), including 58 LTS (36% mutated). No macroscopic residual disease was observed in 58.4% patients. First-line platinum-based chemotherapy plus paclitaxel was administered in 80.4% of patients (median: six cycles (range: 1–14)). After a 9 point 3-year follow-up, median OS was four years (95% CI: 3.6–4.5). The CPS at five years after surviving one year was 42.8% (95% CI: 35.3–48.3); it increased to 81.7% (95% CI: 75.5–87.8) after four survived years. Progression-free interval>18 months was the only LTS prognostic factor in the multivariable analysis (hazard ratio (HR) = 0.23; 95% CI: 0.13–0.40; *p* < 0.001). *Conclusion:* The CPS provided relevant and encouraging clinical information on the life expectancy of HGSOC patients who already survived a period of time after diagnosis. LTS prognostic factors are useful for clinicians and patients.

## 1. Introduction

High-grade serous ovarian cancers (HGSOC) are heterogeneous [1] and lack specific clinical signs and efficient screening [2]. They are often diagnosed at an advanced stage and associated with poor survival rates, i.e., 39% of the five-year overall survival (OS) rate [3]. According the French Registry, the probability of OS at five years reaches 41% for patients treated for ovarian cancer in the 1989–2010 period [4]. The current therapeutic strategy combines maximal cytoreductive surgery and platinum-based chemotherapy [5,6]. Despite an initial sensitivity to cytostatic drugs, 75% of patients eventually relapse at 12–18 months [3,5,7]. The clinical prognostic factors associated with OS included age, stage (I–II vs. III–IV) performance status, and low-grade histology. The role of the residual disease after surgery remains essential for impacting the future of the patients who could be operated [8,9]. More recently, for patients who are not eligible for initial macroscopic complete resection and treated with neoadjuvant chemotherapy, the addition of hyperthermic intraperitoneal chemotherapy has demonstrated a 12-month increase in OS [10]. As biological prognostic factors, the perioperative kinetics of CA 125 has been found to influence the OS [11]. Moreover, patients with BRCA mutations have been shown to survive longer than others and harbor higher intrinsic platinum sensitivity [12], although, for other authors, BRCA mutations alone cannot explain the long-term survival [13].

For HGSOC patients, death usually occurs within 36 to 50 months after diagnosis [7]. Around 15% to 20% of patients are considered long-term survivors (LTS) once they have survived more than five or six years after diagnosis, even considering the advanced stage of disease [12,14]. The definition of LTS varies regarding staging, grading, histology, and biology [15]. Since the median of survival reaches around 50 months, the vast majority of the authors consider LTS beyond this median. We choose 60 months in order to observe a clear decrease of the probability to die of the disease [15]. Regardless of the surgical or oncologic treatments they have received, this specific LTS population is expected to harbor particular biological characteristics such as germline or somatic homologous recombination deficiency.

The prognosis factors of LTS are not clearly defined, although these are needed in clinical practice both for clinicians and patients. Published prognostic factors for patients’ survival provide essential information for estimating their survival rate at diagnosis. However, for patients who survive more than five years without a relapse, these factors are less instructive for evaluating the likelihood of a late recurrence and death from the disease. The evolution of ovarian cancer is often described in terms of survival rates, calculated from the date of cancer diagnosis and estimated by the Kaplan-Meier method. For those patients who have already survived a period of time after diagnosis, the question is whether their life expectancy can be readjusted considering their already quite long survival. Moreover, as death events occur in a nonlinear model, readjusting the probability of survival according to the already survived interval from diagnosis is justified. The conditional probability of survival, i.e., the probability of survival for a patient who has already survived a period of time after diagnosis and treatment, offers these patients more accurate and relevant information. This has been assessed for many different cancer sites (lung [16,17], digestive system [18,19,20], brain [21], head and neck cancers [22], and melanomas [23,24]) and, also, in studies using data from the SEER (Surveillance, Epidemiology and End Results) database [25,26,27].

Two studies assessed [14,25] the conditional survival in patients with ovarian cancer, showing that conditional survival was a more accurate estimation than an estimation made at the time of diagnosis (conventional survival estimation). For a given already survived period of time, the probability of surviving longer increases compared with the initial probability at diagnosis, especially for patients with serous and poorly differentiated tumors. However, these studies did not assess all demographic and new biological tumor factors. Indeed, the standards of care in ovarian cancer have evolved, especially with the incorporation of poly(ADP-ribose) polymérase (PARP) inhibitors for maintenance therapy as a first line of treatment or for recurrence [28,29,30,31,32,33,34].

In this context, we conducted a retrospective study of HGSOC LTS patients, i.e., patients who had already survived five years after diagnosis, to estimate the conditional probability of their survival and identify the prognostic factors of the OS.

## 2. Results

### 2.1. Patients

Among the 553 patients recorded in the ovarian cancer database, 404 were analyzed in the study, including 120 patients (29.7%) considered as long-term survivors (LTS) (Table 1). The median age at diagnosis was 57 years (range: 21–83) for LTS and 62 years (range: 20–89) for the non-LTS. More patients were younger at diagnosis (<65 years old) among LTS than among the non-LTS (75.8% vs. 59.2%, *p* < 0.001). The WHO performance status was 0 to 1 and 2 in 92.5% and 7.5% of the LTS group and 84.7% and 15.3% of the non-LTS group (*p* = 0.045). CA 125 levels and staging at the baseline did not differ for either group. Overall, most patients (82.7%) had a FIGO stage III tumor and 17.3% FIGO stage IV. In the whole population, 32.3% of patients presented visceral metastases. BRCA status was available for 28.7% of all patients. Among them, BRCA 1 and 2 genes were mutated in 36.2% of LTS and in 29.3% of non-LTS, which was not significantly different between the two groups.

### 2.2. Treatment

Primary debulking surgery was performed in 67.5% and 46.8% in the LTS and non-LTS groups (*p* < 0.001) (Table 2). Macroscopic residual disease was null for 63.3% of LTS and 56.3% of non-LTS. A peritoneal carcinomatosis index (PCI) > 17 was reported in 17.9% of LTS vs. 37.0% in the non-LTS group (*p* < 0.001). Neoadjuvant chemotherapy was administered to 31.1% of LTS and 51.9% of the non-LTS (*p* < 0.001), with a median number of cycles of six (range: 1–14) for both groups. Eighty percent of patients were treated with first-line platinum-based chemotherapy plus paclitaxel. Primary platinum sensitivity of the tumors was reported more frequently in the LTS group, with 95% in the LTS group and 68% in the non-LTS group (*p* = 0.001). A treatment-free interval (TFI) ≥ 18 months was reported for 57.8% and 14.4% of patients in the LTS and non-LTS groups (*p* < 0.001), and a progression-free interval (PFI) ≥ 18 months was reported in 85.8% and 33.5% of patients, respectively (*p* < 0.001).

### 2.3. Survival and Conditional Survival

After a median follow-up of nine point three years (range: 0.15–20.8), the median OS for the whole population was four years (95% CI: 3.6–4.5), and the median relapse-free survival was one point sixyears (95% CI: 1.5–1.8). The CPS at five years after surviving one year after diagnosis was 42.8% (95% CI: 35.3–48.3), and it increased up to 50.9% (95% CI: 44.8–57.0), 62% (95% CI: 55.4–68.7), and 81.7% (95% CI: 75.5–87.8) after surviving two, three, and four years, respectively (Table 3). Conditional survival at five years and the CPS for one additional year according to the time since diagnosis are reported in Figure 1a,b. Figure 1b shows *n +* oneyear CPS estimates for patients who have already survived *n* years, in which *n* varies from zero to six. CPS for one additional year decreases until four years after diagnosis, then seems to stabilize subsequently.

### 2.4. Prognostic Factors of Long-Term Survival According to the Five-Year Landmark Analysis

Prognostic factors for LTS were assessed using the Landmark analysis at five years in a univariate analysis. Age < 65 years, complete surgery, initial PCI ≤ 17, and PFI ≥ 18 months were found to be significant factors associated with a good prognosis (Table 4). In the multivariate analysis, only the progression-free interval was found to be significant (HR = 0.23; 95% CI: 0.13–0.40; *p* < 0.001).

## 3. Discussion

In our population of HGSOC patients, the probability of surviving after having already survived a given time after diagnosis increased in the first two to three years after diagnosis, then stabilized from the fifth year and beyond. We showed that the probability of surviving one additional year after four years post-diagnosis was 81.7% (95% CI: 75.5; 87.8), while the classical five-year OS was 40.6% (95% CI: 35.3; 45.9). The CPS for one additional year first decreased, then seemed to stabilize after four years, suggesting a more optimistic future. Our results confirm that CPS is a precious tool for physicians to use during their patients′ follow-ups and to inform them of their life expectancy. It is indeed one of the greatest expectations from patients to hear probable survival figures, allowing them to make projects more easily in their personal or professional lives.

Two previous studies analyzed the CPS in ovarian cancer patients. A first study from the SEER database showed that, in patients treated between 1988 and 2001, the five-year OS improved in patients who had already survived five years [25]. However, these results are to be read with caution, as standards of practice have evolved since then, especially with the development of platinum-based and taxane-based chemotherapies. The prognosis is now better than at the time of that study. Another study confirmed these data on CPS. They showed that, in Stage I to IV ovarian cancer patients, the conditional disease-free survival increased over time, even in patients with an initial high risk of recurrence [14]. However, this study included a fairly heterogeneous population compared with other studies (including ours, because our patients were only Stage III-IV HGSOC). Their endpoint was to study CPS based on PFS, unlike OS in ours. The median OS was four point five years and four years in Kurta’s study and our study, respectively. It is notable that, at that period of time, our population was not receiving any maintenance treatment with a significant impact, such as PARP inhibitors. Despite this fact, we wish to point out the heterogeneity of high-stage HGSOC behaviors beyond macroscopic complete resection.

The second objective of our study was to identify additional prognostic factors of long-term survival, especially biological parameters that might characterize subgroups of LTS. Our results in the univariate analysis were in accordance with the standard prognostic factors usually reported, i.e., age, absence of a macroscopic residual disease after primary or interval cytoreductive surgery, and the extent of peritoneal disease (PCI) [8,9,34,35,36]. In previous studies, many attempts to identify a PCI threshold as a prognostic factor have been performed. The scores vary between 13 to 17 [37,38]. We choose 17 as the cutoff according to what was published in colorectal and ovarian carcinomatosis. In an attempt to definitively sterilize macroscopic disease, intraperitoneal chemotherapy has been used after primary debulking surgery or HIPEC in interval surgery, with results to be confirmed.

In the multivariate analysis, only the PFI was found to be a prognostic factor for LTS; having survived more than 18 months between the first line of chemotherapy and the date of recurrence was considered to favorably impact the OS with a hazard ratio reaching 0.23 (95% CI: 0.13–0.40) in a Landmark analysis at five years. In other terms, the longer the disease is controlled, the greater the likelihood of a definitive cure. This effect has also been reported for other high-grade malignant diseases [21].

Sensitivity of the disease to platinum compounds, defined as a delay in recurrence of more than six months after the first line of treatment, is also a strong indicator of long-term survival, since 95% of the LTS patients in our study were sensitive to platinum-based treatments. If we consider that platinum sensitivity is correlated with the deficit of homologous recombination mediated mainly by BRCA 1 and 2, and also other genes arising in 51% of the cases, these genes can be expected to represent markers for LTS and to become a major surrogate factor for the prognostic assessment and help in adapting the subsequent appropriate treatment. In our study, as we had too-few patients with a known BRCA status, we were unable to show either a better prognosis [12], as already demonstrated for mutated patients, or a higher frequency of mutated patients among the long-term survivors. Prospects include conducting a specific study to better characterize the biological factors that are predictive of platinum sensitivity, such as factors involved in homologous recombination, tumor microenvironment (PD-L1 status and immune infiltrate), or immunoreaction [39], for which a first set of results will be soon available. Another limitation of the study is the relatively short follow-up time, considering the context and population studied.

## 4. Patients and Methods

### 4.1. Database

The ovarian cancer database at our institution was created in 1995 and has been regularly implemented since then. It currently holds information on around 600 patients with high-grade epithelial ovarian cancer. For study purposes, the date of diagnosis; tumor histology; FIGO staging; peritoneal carcinomatosis index (PCI); detailed treatment data (primary or interval debulking surgery, residual macroscopic disease, neoadjuvant or adjuvant chemotherapy, type of chemotherapy, and number of cycles); BRCA 1 and 2 status; recurrence information; duration of treatment-free interval; and date of death or date and condition at the last follow-up were all recorded.

### 4.2. Study Patients

Study data were extracted from the ovarian cancer database at our institution. All consecutive patients aged 18 or over, treated at our institution for FIGO stage III -IV HGSOC since 1995, were included. The study was conducted in accordance with Good Clinical Practices, the Helsinki Declaration and was approved by the Montpellier Cancer Institute Institutional Review Board (ID number ICM-CORT-2016-09). All patients gave their written, informed consent.

### 4.3. Treatment

Patients considered as eligible for primary debulking surgery were operated, then received 6 to 9 cycles of combined platinum-based chemotherapy in addition. Patients with extended disease who were not eligible for primary debulking surgery received 3 to 4 cycles of neoadjuvant chemotherapy and then underwent interval debulking surgery followed by 3 to 6 cycles of the same treatment. None of the patients received first-line treatments with bevacizumab, PARP inhibitors, or immunotherapy.

### 4.4. Statistical Considerations

Descriptive analyses are reported with medians and ranges for continuous variables and frequencies and percentages for categorical variables. The patients’ characteristics were compared using the Kruskal-Wallis or Wilcoxon tests for continuous variables, and the chi-square or Fisher’s exact tests for categorical variables. The treatment-free interval (TFI) was defined as the time from the end date of the 1st chemotherapy to the start date of the 2nd chemotherapy and the progression-free interval (PFI) as the time from the start date of the 1st chemotherapy to the date of progression. Note that the TFI is missing for the patients who only received the 1st line of chemotherapy.

The primary endpoint was the OS, defined as the time from the date of diagnosis to the date of last contact or documented death from any cause. OS rates (and their 95% confidence intervals, 95% CI) were estimated using the Kaplan Meier method and the median follow-up using the reverse Kaplan-Meier method.

The conditional probability of survival (CPS) was the probability of surviving Y years after diagnosis when the patient had already survived X years (Y > X). For example, to compute the 5-year CPS estimates for patients who had already survived 1 year, the 5-year OS rate was divided by the 1-year OS rate. Ninety-five percent CIs were estimated using a variation of the Greenwood formula [40].

To identify prognostic factors for OS in LTS, the Landmark analysis approach was used. We focused on patients who had already survived at least 5 years after diagnosis. Patients who died or were censored before the Landmark time were excluded from analysis. The Cox proportional hazard model was then used to determine the prognostic factors for LTS. All significance tests were two-sided, and statistical significance was set at *p* < 0.05. All analyses were conducted using Stata software version 13 (StataCorp LP, College Station, TX, USA).

## 5. Conclusions

In conclusion, to the best of our knowledge, this study based on a homogeneous population of HGSOC patients is the largest series used for a retrospective analysis. It confirms that CPS can provide relevant, encouraging clinical information about the remaining life expectancy of ovarian cancer patients who have already survived a certain period of time since the diagnosis.

## Figures and Tables

**Figure 1 cancers-12-02184-f001:**
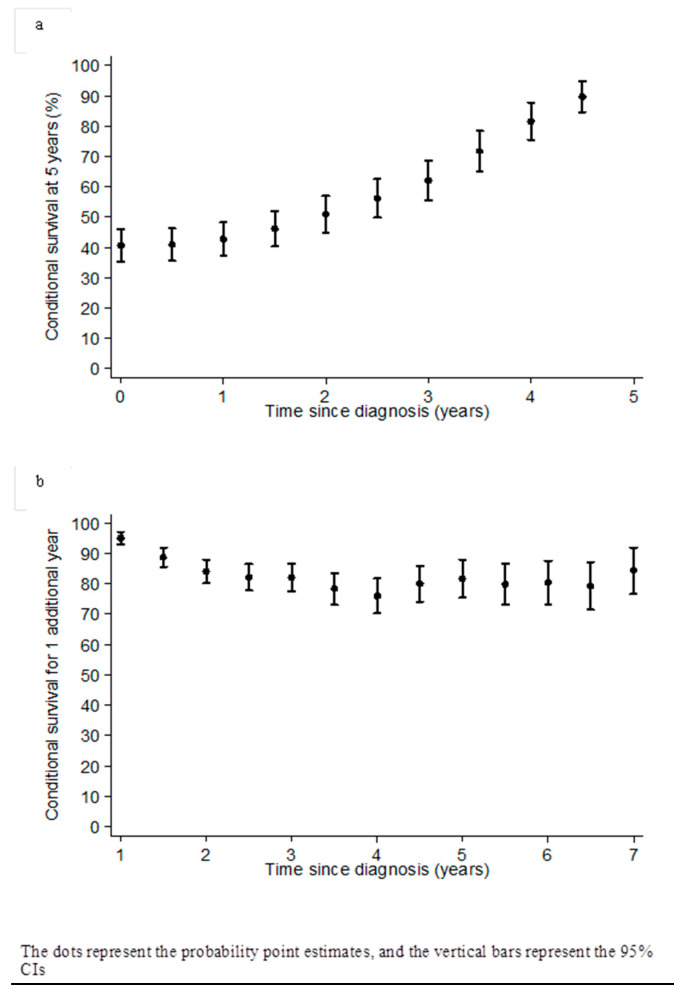
Conditional probability of survival at 5 years (**a**) and conditional probability of survival for one additional year (**b**) according to the time of diagnosis

**Table 1 cancers-12-02184-t001:** Patients′ characteristics at baseline.

Characteristics	Non-Long Survivors *n* = 284		Long-Term Survivors *n* = 120		Total *n* = 404		*p*-Value
	N	%	N	%	N	%	
**Age**							
Median (range)	62 (20–89)		57 (21–83)		61 (20–89)		<0.001
<65	168	59.2	91	75.8	259	64.1	0.001
≥65	116	40.8	29	24.2	145	35.9	
**WHO Performance status**							0.045
0–1	227	84.7	98	92.5	325	86.9	
2	41	15.3	8	7.5	49	13.1	
Missing	16		14		30		
**CA 125 level**							
Median (range)	700 (6–21)		790 (14–12)		734 (6–22)		0.360
<70	16	7.5	4	4.9	20	6.8	0.434
≥70	197	92.5	77	95.1	274	93.2	
Missing	71		39		110		
**FIGO stage**							0.168
III	230	81.0	104	86.7	334	82.7	
IV	54	19.0	16	13.3	70	17.3	
**Metastasis localization (*n* = 70)**							
Pleural	34	69.4	8	61.5	42	67.7	
Visceral	15	30.6	5	38.5	20	32.3	
Missing	5		3		8		
N stage							0.431
N−	63	28.0	32	32.3	95	29.3	
N+	162	72.0	67	67.7	229	70.7	
Missing	59		21		80		
**BRCA mutation**							0.768
Mutated BRCA1	12	20.7	15	25.9	27	23.3	
Mutated BRCA2	5	8.6	6	10.3	11	9.5	
No mutation	41	70.7	37	63.8	78	67.2	
Missing	226		62		288		

WHO: World Health Organization, FIGO: Fédération Internationale de Gynécologie et d’Obstétrique, BRCA: breast cancer.

**Table 2 cancers-12-02184-t002:** Treatments.

Characteristics	Non-Long Survivors *n* = 284		Long-Term Survivors *n* = 120		Total *n* = 404		*p*-Value
	N	%	N	%	N	%	
**Surgery type**							<0.001
Primary	133	46.8	81	67.5	214	53.0	
Interval surgery	151	53.2	39	32.5	190	47.0	
**Digestive resection**							<0.001
0	87	43.1	66	60.0	153	49.0	
>1	115	56.9	44	40.0	159	51.0	
Missing	82		10		92		
**Splenectomy**							0.241
No	128	96.2	75	92.6	203	94.9	
Yes	5	3.8	6	7.4	11	5.1	
Missing	151		39		190		
**Primary or interval surgery**							0.192
Complete (Gross total resection)	160	56.3	76	63.3	236	58.4	
Incomplete (Macroscopic residual disease)	124	43.7	44	36.7	168	41.6	
**Initial PCI**							
Median (range)	14 (0–33)		11 (0–24)		13 (0–33)		<0.001
<17	165	63.0	87	82.1	252	68.5	<0.001
≥17	97	37.0	19	17.9	116	31.5	
Missing	22		14		36		
**Chemotherapy**							
Neoadjuvant CT	147	51.9	37	31.1	184	45.8	<0.001
Missing	1		1		2		
**1st CT line regimen**							0.793
Platinum salts/Paclitaxel	223	79.7	97	82.2	320	80.4	
Platinum salts/Exoxan	30	10.7	12	10.2	42	10.6	
Others	27	9.6	9	7.6	36	9.0	
Missing	4		2		6		
**Number of cycles**							0.946
Median (range)	6 (1–14)		6 (3–12)		6 (1–14)		
Missing	6		1		7		
**Sensibility to 1st CT line**							0.001
Refractory/Resistant	85	31.8	6	5.0	91	23.5	
Sensible	181	68.2	114	95.0	295	76.5	
Missing	18		0		18		
**Treatment-free Interval (months)**							<0.001
Median (range)	8.5 (0.4–47.2)		23.9 (0.7–121.6)		11 (0.4–121.6)		
<18	149	85.6	35	42.2	184	71.6	
≥18	25	14.4	48	57.8	73	28.4	
Missing/No 2nd CT line	110		37		147		
**Progression-free Interval (months)**							<0.001
Median (range)	14.5 (0.4–56.6)		43.4 (2.3–248.2)		17.4 (0.4–248.2)		
<18	187	66.6	17	14.2	204	50.9	
≥18	94	33.5	103	85.8	197	49.1	
Missing	3		0		3		

PCI: peritoneal carcinomatosis index. CT: chemotherapy.

**Table 3 cancers-12-02184-t003:** Overall survival rates (left) and conditional probabilities of survival (right) depending on the survived time interval.

*n* = 404											
Time	Overall Survival Rate			Conditional Probability of Survival							
				2 Years		3 Years		4 Years		5 Years	
	%	95% CI	No. at risk	%	95% CI	%	95% CI	%	95% CI	%	95% CI
1 year	95.0	(92.4; 96.7)	373	84.0	(80.2; 87.8)	68.9	(64.0; 73.8)	52.4	(46.9; 57.8)	42.8	(37.3; 48.3)
2 years	79.8	(75.4; 83.5)	279			82.0	(77.4; 86.6)	62.3	(56.5; 68.2)	50.9	(44.8; 57.0)
3 years	65.5	(60.3; 70.1)	219					76.0	(70.2; 81.7)	62.0	(55.4; 68.7)
4 years	49.7	(44.3; 54.9)	158							81.7	(75.5; 87.8)
5 years	40.6	(35.3; 45.9)	120								

**Table 4 cancers-12-02184-t004:** Prognostic factors of long survival in univariate and multivariate analyses.

*n* = 120	Univariate Analysis		Multivariate Analysis	
	HR	95% CI	HR	95% CI
**Age**				
<65	1			
≥65	1.60	(0.95; 2.70)		
	*p* = 0.085			
**WHO Performance status**				
0	1			
1–2	1.15	(0.42; 3.18)		
	*p* = 0.790			
**FIGO stage**				
III	1			
IV	1.08	(0.53; 2.19)		
	*p* = 0.830			
N stage				
N−	1			
N+	1.33	(0.75; 2.38)		
	*p* = 0.325			
BRCA mutation				
Mutated BRCA1/2	1			
No mutation	1.02	(0.43; 2.44)		
	*p* = 0.958			
Primary or interval surgery				
Complete	1			
Incomplete	1.92	(1.17; 3.14)		
	*p* = 0.011			
Initial PCI				
<17	1			
≥17	1.78	(0.96; 3.27)		
	*p* = 0.079			
Sensibility to 1st CT line				
Refractory/Resistant	1			
Sensitive	0.47	(0.19; 1.19)		
	*p* = 0.149			
Progression-free interval				
<18 months	1		1	
≥18 months	0.23	(0.13; 0.40)	0.23	[0.13; 0.40]
	*p* < 0.0001		*p* < 0.0001	

HR: hazard ratio, 95% CI: 95% confidence interval, WHO: World Health Organization, FIGO: Fédération Internationale de Gynécologie et d’Obstétrique, BRCA1: breast cancer 1, PCI: peritoneal carcinomatosis index, and CT: chemotherapy.

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
