# Peer review of "Conditional Probability of Survival and Prognostic Factors in Long-Term Survivors of High-Grade Serous Ovarian Cancer"

_cancers, 2020, doi:10.3390/cancers12082184_

Round 1

Reviewer 1 Report

In general, the study is very interesting and up to date. There is no evident bias. However, I feel that the authors could provide additional analysis to fulfill the study.

Please provide clear definition for LTS you have chosen for the study. It is not clearly written.

You should divide FIGO st III patient into subgroups. There is clear difference in survival between IIIa and IIIC patients.

What does it mean “complete” or “incomplete” surgery? In general, in ovarian cancer surgery we prefer to divide patients into three groups: 1) no gross residual disease (CC=0 acc. To Sugarbaker, or maximal cytoreduction, or R0 resection), 2) optimal cytoreduction (tumor less than 1 cm, R =1), 3) nonoptimal cytoreduction (tumors above 1 cm, gross residual bulky, R =2). For the first, please provide adequate caption for the table. Additionally, it would be also interesting, if you could divide the study group according to 3 subgroups regarding the residual disease.  

Why did you chose PCI 17 as a cut off? Did you analyzed different cut-offs? Try to performe ROC analysis.

The authors have performed the analysis of long term outcome in relation to the overall survival. However, if we focus for example of patients who survived 3 years, there are the groups of patients who are tumor free, and the patients who are treated due to cancer relapse. Therefore, it would be interesting, to separate tumor free and relapsed patients in the analysis.

There is lack of data concerning surgical treatment. The residual disease following surgery is regarded as the strongest prognostic factor for ovarian cancer. But on the other hand, the survival is negatively correlated with the extent of the surgery. Is it possible to include the range of the surgery into the analysis? For example to divide IIIc patients into the groups: who required bowel resection vs no-bowel resection? If not, please provide detailed information about surgical treatment you performed? Did you perform bowel resection? Liver metastases? Splenectomy?

What type of criteria did you use for the qualification for interval debulking surgery? I would also recommend, to perform additional evaluation for PDS and IDS patients.

Reviewer 2 Report

1. The introduction should have more concept of the factors that related to the prognosis of HGSOC patients.
2. What is the prognosis of ovarian cancer in your country; this should be mentioned inside the article.
3. What limitations of retrospective design should be described more clearly?
4. The cause of death in both groups should be analyzed more clearly, the complication of cancer, or other comorbidity should also be involved.
Some of the chronic disease or comorbidity that would also cause death, or hold the treatment should also be considered in the prognostic factors, like poor renal function or liver function.

Reviewer 3 Report

This is the review of a manuscript by Fabbro et al concerning conditional prediction of the survival in patients with high grade serous ovarian carcinoma beyond 5 year horizon. 

This paper is of good quality, it follows in footsteps of two earlier analysis, one performed using SEER database (now this analysis is in a larger part obsolete due to novel treatment modalities), and another by Kurta, published in 2014, where patients were recruited as part of the Hormones and Ovarian Cancer Prediction case-control study (N=404)

Current study does not exceed the population size of Kurta et al (oddly, it also retrospectively enrolled 404 patients), but differ from one by Kurta as it provides a window of overall survival instead of progression free survival.  

Overall, the manuscript is well written, a set of statistical tool employed in the study is standard, the procedures are executed correctly, and the conclusion is meaningful, both for patients and for treating physicians. Results are well-presented. 

Overall, i have no problems with this study, especially as it seems that the study was already previously peer-reviewed and the suggested augmenting changes implements. I base my conclusion on the paragraph concerning analysis of the mutations as additional predictors (not powerful enough). 

On a minor note, I would like authors to consider inclusion in the Discussion some phrases about PCI scores which are verging on becoming significant if the population would be larger. 

Overall, good manuscript. 
